# Effect of Ending the Nationwide Free School Fruit Scheme on the Intake of Fruits, Vegetables, and Unhealthy Snacks in Norwegian School Children Aged 10–12 Years

**DOI:** 10.3390/ijerph20032489

**Published:** 2023-01-30

**Authors:** Helene Kristin Richardsen, Elling Tufte Bere, Tonje Holte Stea, Knut-Inge Klepp, Dagrun Engeset

**Affiliations:** 1Department of Nutrition and Public Health, University of Agder, 4630 Kristiansand, Norway; 2Department of Sport Science and Physical Education, University of Agder, 4630 Kristiansand, Norway; 3Department of Health and Inequality, Norwegian Institute of Public Health, 0213 Oslo, Norway; 4Department of Health and Nursing Sciences, University of Agder, 4630 Kristiansand, Norway; 5Division of Mental and Physical Health, Norwegian Institute of Public Health, 0213 Oslo, Norway; 6Department of Nutrition, Faculty of Medicine, University of Oslo, 0315 Oslo, Norway

**Keywords:** school children, withdrawal, fruit and vegetable intake, school fruit program, Norway

## Abstract

The Norwegian authorities started a nationwide free school fruit program in 2007, implemented in all secondary schools (grades 8–10) and combined schools (grades 1–10) in Norway. The program ended in 2014. This study evaluates the effect of ending the nationwide free school fruit program on the consumption of fruit, vegetables, and unhealthy snacks among Norwegian sixth and seventh graders. The study sample consists of pupils at 18 schools that participated in all data collections in the Fruits and Vegetables Make the Marks project (FVMM), initiated in 2001, with new data collections in 2008 and 2018. Four of the schools were combined schools, therefore children in sixth and seventh grade at these schools received free fruit in 2008 (intervention schools), and fourteen schools did not (control schools). Between 2008 and 2018, pupils at the intervention schools ate a lower proportion of fruits and vegetables per school week, and the consumption of unhealthy snacks increased compared to the control schools. Completion of the free fruit program was not significantly different for boys and girls, or low and high parental education. The results indicate that the end of the free school fruit program resulted in less healthy eating habits among children.

## 1. Introduction

Childhood and adolescence are considered a critical period due to increased nutritional requirements, including total energy intake, to meet the body’s physiological needs [1,2].

Globally, an unhealthy diet seems to be the largest risk factor for non-communicable diseases (NCD) [3,4], and a low consumption of fruits and vegetables (FV) among children and adolescents is considered as one of the leading dietary risk factors among large population groups from different countries and continents [5]. The WHO encourages all countries and schools to prioritize healthy food in schools, since the school is an arena where many different people can be reached, and thus influenced to adopt a healthier diet [6]. The EU Parliament and Council established a school scheme for fruit and vegetable supply in schools (Regulation (EU) No 1308/2013) [7]. Several evaluations of the school scheme has been conducted, both within individual member states [8,9] and among all EU member states [10]. They all concluded that the school scheme could be an effective strategy for increasing the FV consumption among children [8,9,10]. In the US, a Fresh Fruit and Vegetable Program (FFVP) has been implemented at low-income schools, and the results from the program has shown a higher intake of FV after school hours among children participating in the program [11].

Several studies have indicated socioeconomic differences in dietary habits, including FV intake, as people with higher education usually eat more FV than those with lower education [12,13]. In addition, health promotion measures are often more effective among families with a high socioeconomic background, which may contribute to an increase rather than a decrease in health disparities among different socioeconomic groups [14]. However, providing free school meals seems to be one of few effective initiatives contributing to the reduction of the socioeconomic gap [15,16,17]. The importance of free school meals is also emphasized by results from tracking studies indicating that dietary habits established in childhood/adolescence tend to be maintained in adulthood. [15,18,19,20]. 

In Norway, low intake of fruit and vegetables and high intake of salt are considered the leading dietary risk factors for mortality among both sexes [21]. The Norwegian Directorate of Health recommends a daily intake of at least 250 g of fruit and 250 g of vegetables for adults and children over the age of 10. For younger children, there is no specific quantity recommendation, but a rule of thumb has been five servings the size of the child’s handful [22]. Norwegian children consume about half of the recommended intake of fruit and vegetables (FV), and more than the recommended intake of added sugar (which is less than 10 percent of total energy intake) [23]. Thus, increased FV intake among children has become an important objective in the public health policy, and in Norway, there has been a political agreement that early intervention is crucial for dealing with the health-related social inequalities [24,25,26,27]. 

Norwegian health authorities have previously made an effort to increase the FV intake among school children as a strategy to reduce social inequalities in health [25]. A nationwide free school fruit program, financed by the state, was implemented in all lower secondary schools (grades 8–10) and combined schools (grades 1–10) in Norway, autumn 2007 [28]. 

An ongoing subscription program with parental payment was initiated in 1996 (grades 1–7) and made nationwide in 2003 (for grades 1–10), in collaboration with the Norwegian Marketing Board for Fruit and Vegetables [29]. The program is subsidised by the Norwegian government, and the individual school decides for itself whether they want to participate in the program, and it is voluntary for parents to make use of the offer [29].

The subscription program and a pilot version of the free school fruit program (without parental payment) were evaluated by the Fruits and Vegetables Make the Marks (FVMM) project [30]. A school-randomised trial with a cohort of 1950 pupils from thirty-eight schools (sixth and seventh grades) was followed for three years, from 2001/2002 to 2004/2005. The results showed an increased FV intake for both programs, but that the free school fruit program was most effective [30].

An evaluation of the nationwide free school fruit program also showed an increase in pupils’ intake of FV at school between 2001 and 2008, with the largest increase for schools included in the free school program and smallest for schools with neither free nor subscription program [29,31]. Despite results indicating positive effects of the free school fruit program, the initiative was terminated by the national government in 2014 [32]. 

Studies of discontinuation of interventions may expand the possibility of evaluating interventions and adding to the evidence base [33], but to our knowledge no previous study has examined the effects of ending a free school fruit program. Ending the state free fruit intervention thus provided an excellent opportunity to evaluate the effect of this using a natural experimental design, which can provide useful knowledge for future public health policy and research. This study aimed to evaluate the effect terminating the nationwide free school fruit program had on children’s (10–12 years) consumption of FV and unhealthy snacks. The effect will be assessed in relation to socioeconomic status and gender.

## 2. Methods

### 2.1. Study Design and Sample

In the 2001–2002 school year, a subscription program for fruits and vegetables was about to start in the counties Hedmark and Telemark; therefore, these counties were selected for carrying out the FVMM project [34]. Both Hedmark and Telemark are situated in the south-east of Norway and are considered similar regarding socioeconomic composition and geography (Figure 1). 

The study was conducted among primary school children from the two counties, and 48 schools were randomly selected and invited to participate. Nineteen schools from Telemark and nineteen schools from Hedmark agreed to participate in the study. A follow-up study was conducted in 2008, and a second follow-up was conducted in 2018. Eighteen schools—ten schools from Hedmark and eight schools from Telemark—participated both at baseline and in the two follow-up studies and constitute the sample of the present study (Figure 2). 

The Free Fruit program was primarily intended for all Norwegian lower secondary schools (grades 8–10). However, four of the schools in this sample were combined schools (grades 1–10) and, therefore, all pupils at these schools received the nationwide free fruit program. Sixth and seventh graders (age 10–12 years) from the combined schools represent the intervention group. The primary schools (grades 1–7) (control group) did not receive free fruit, but some of them had a subscription program. 

The free school fruit program constitutes a quasi-experiment, since treatment assessment is a result of the national school fruit policies, rather than being controlled in the traditional sense of a randomized trial. 

### 2.2. The Questionnaires

The FVMM project used two questionnaires, one that was designed and validated for pupils aged 10–12 years and one for the pupils’ parents [35]. Both questionnaires included a modified 24 h recall, a food frequency questionnaire (FFQ), questions assessing attitudes towards FV, demographic questions, and questions concerning other health-related behaviors.

Pupils’ questionnaire: The modified 24 h recall was divided into five periods/meals throughout the day (before school/breakfast, at school/lunch, after school, dinner, and in the evening/supper) to make it easier for the children to remember. The children were told to record all fruits, berries, and vegetables measured in number (e.g., one apple, and one banana) or in portion (e.g., a portion of fruit salad) for the previous day. One portion was set at about 80 g (ranging from 65 g (one carrot, etc.) to 105 g (one apple/one orange)). The portion of fruits from each period of the day was added together, making a fruit score (portions/day). The same was done for vegetables, making a vegetable score, and fruits and vegetables were summed together in a combined FV score [35]. The conversions from household measures to portions were based on the report “Weights, measures and portion sizes for foods” [36]. 

In addition to the fruit, vegetable, and FV score, the question “How often do you eat fruits and/or vegetables at school?” from the FFQ was included in the analyses. The response alternatives were «every school day, 4, 3, 2, and 1 d/week, less than once a week, never, and don’t know». The FFQ was dichotomized into eating FV at school 4–5 d/week or less than 4 d/week, indicating eating FV at most school days. The 24 h recall and the FFQ have been validated, and a test–retest study shows consistent reliability among 6th graders [35]. A validation study of self-reported FV intake showed that 6th graders were capable of recording yesterday’s intake of vegetables but overestimated the intake of fruit. The ability to rank subjects based on the FFQ was rather low, but equal to similar studies [35]. 

A sum score of unhealthy snacks was made from the following 3 items: “How often do you drink sugar-sweetened soft drink?”, “How often do you eat potato chips?”, and “How often do you eat candy (chocolate, mixed candy, etc.)?”. All items had 10 response alternatives and were scored as follows—never (0), less than once a week (0.5), once a week (1), twice a week (2), …, 6 times a week (6), every day (7), several times every day (10)—giving the unhealthy snacks scale a range from 0 to 30 times/week. The unhealthy snack score showed good reliability in a test–retest by sixth graders 14 days apart [35].

A trained project worker read out loud the questionnaire used for the children, and the questionnaire was completed by the children in their classroom. About 45 min were used to complete the questionnaire. All the pupils were tested on weekdays (Tuesday to Friday). 

Parents answered a similar questionnaire to the pupil questionnaire but a question about their education level was included. It was reported with the following response options: elementary school, high school, university college or university (three years or less), and university college or university (more than three years). This was later dichotomized into lower (no college or university education) and higher (college or university education) level of education [37]. In the present study, only the variables about parental education level were used as a measure for socioeconomic status. 

### 2.3. Statistical Analysis

For descriptive analyses, independent-sample t-tests were conducted for continuous variables, and chi-square for dichotomous variables for pairwise comparisons. The main analyses were multi-level mixed models with fruit, vegetables, and FV at school and throughout the day, and unhealthy snacks as separate outcome variables. Year (2008 and 2018), group, sex, and parental education level were used as fixed effects, and school was used as a random effect. A significant time × group interaction (*p* < 0.1), indicating different changes in consumption over time for the intervention group (had free fruit program in 2008) and the control group (had no program or subscription program in 2008), was used to test the effect of ending the school fruit program. To assess potential differences in the effect of ending the school fruit program for different groups (based on sex and parental education level), the third-order interaction time × group × sex, and time × group × parental education level was examined. An examination of the residuals of the continuous variables did not reveal unacceptable departures from normality.

Pupils that did not attend school the day before the survey day (28 in 2001, 19 in 2008, and 13 in 2018) were excluded from the analyses of intake of FV at school but included in all other analyses. 

To conduct a school attrition analysis, pupils at the 18 schools in the present study sample were pairwise compared at baseline with those at the 20 schools that did not participate in 2008 or 2018, regarding all variables. All analyses were conducted using the Statistical Package of Social Sciences (SPSS), version 24 (SPSS Inc., Chicago, IL, USA). The significance level was set at *p* < 0.05. 

### 2.4. Ethic

Informed consent was signed by the parents prior to each study and collected by a contact person at the schools. The present study was conducted according to the guidelines laid down in the Declaration of Helsinki, and all procedures involving human subjects were approved by the Norwegian Center of Research Data for all time points and The National Committees for Research Ethics in Norway in 2001. 

## 3. Results

A total of 1472 pupils from the 18 schools constituting this survey completed the questionnaire in 2008 and 2018. In 2008, the participation rate was much higher than in 2018 (78% and 44%, respectively, for the 27 schools participating in 2008 and 25 schools from 2018) (Table 1). A total of 668 and 431 parents answered the parental questionnaire in 2008 and 2018, respectively, of which 79% of the respondents were mothers at both time points. 

All-day FV intake for the sixth and seventh grade pupils at the eighteen schools increased from 2.4 to 3.2 portions/d between 2001 to 2008 and decreased from 3.2 to 2.6 portions between 2008 to 2018, and the proportion reporting to eat FV four or five times/week at school increased from 28% in 2001 to 65% in 2008, with a decrease to 56% in 2018 (*p* = 0.002). In addition, a decrease in intake of unhealthy snacks from 6.9 to 4.6 times/week between 2001 to 2008 (*p* < 0.001) was seen, with a further decrease between 2008 to 2018 (from 4.6 to 4.3 times/week, *p* = 0.016) (Table 1). 

The analysis of fruit and vegetable intake between 2008 and 2018 showed that the time × group interaction was significant for the intake of fruits at school (*p* = 0.047), but not for vegetables and FV combined. The decrease in fruit intake at school was −0.27 and 0.01 portions/day, respectively, for the intervention group and the control group. No significant interaction was observed for the intake of fruits, vegetables, or FV all day. Time × group interaction was also significant for eating FV four or five days/school week (*p* < 0.001), and the consumption of unhealthy snacks (*p* = 0.012). There was a lower proportion (35 percentage points) of pupils in the intervention group eating FV most school days in 2018 than in 2008, compared with the control group where the change was −2 percentage points. The biggest difference was found in the consumption of unhealthy snacks, where the intervention group increased their intake by 0.87 times/week, and the control group decreased their intake by −0.50 times/week (Table 2). 

No significant third-order interactions (time × group × sex, or time × group × parental education level) were observed for any outcome variables, indicating that the effect of ending the free fruit program was not significantly different for boys and girls or low and high parental education. 

The 95% confidence intervals (CI) in Table 2 show that there is an overlap of the CIs for the two groups, which means that there is no statistically significant difference between the means, except for unhealthy snacks in 2008. There is, however, a clear decrease in the consumption of FV and fruit in the intervention group between 2008 and 2018, but not in the control group. For unhealthy snacks, there is a clear increase in consumption between 2008 and 2018 for the intervention group and a decreased consumption for the control group.

Table 3 shows the unadjusted percentage and confidence interval of pupils reporting to eat FV 4 or 5 days per school day stratified into groups. The table reveals no differences in the change between 2008 to 2018 between boys and girls, and low and high parental education for either the intervention schools or the control schools. 

In the attrition analysis, comparing the 18 schools included in the present study sample with the 20 schools that did not participate in 2008 or 2018, the results revealed little differences at baseline. The only difference showed that among the schools not participating at all test points, 34% reported eating FV at school 4 or 5 days a week whereas 28% reported the same among the present study sample (*p* = 0.007). 

## 4. Discussion

This study examined the effect of ending the free school fruit program on the intake of fruits, vegetables, FV, and unhealthy snacks among sixth and seventh graders in two Norwegian counties. The results indicated a negative effect of ending the free fruit program on the intake of fruits at school (−0.27 and 0.01 portions/d change for the intervention schools and control schools, respectively). The decreased intake of FV at school, after the ending of the free fruit program, was higher in the intervention schools than in the control schools. Also, the intervention schools had an increased consumption of unhealthy snacks by almost one portion/d, while the control schools decreased their intake by half a portion. In 2008, however, the intervention schools had a higher intake of FV, and lower intake of unhealthy snacks compared to the control schools. 

One can only speculate on why the intervention group had a larger decrease in FV intake four years after ending the free fruit program, despite a 13-year program period. It might be that the control schools, or the parents in the control schools, made an effort to increase FV intake due to the media focus on FV that followed the free fruit program, and thus, implemented new routines to increase FV intake. The intervention schools, on the other hand, may have gone back to their old routines when the program ended. Ransley et al. [38] suggested similar reasons in their evaluation of the school FV scheme in the north of England. They saw an increased intake of fruits during the intervention, but the intake returned to baseline when the children were no longer part of the FV scheme. An explanation suggested by the authors of the study was that the parents may have stopped providing FV at home because they believed their children were provided with an adequate amount of FV at school [38].

No effect was seen for vegetable intake in the current study. However, this is not surprising as mainly fruit was distributed to participants in the free fruit program and the subscription program. The rationale for not distributing an equal amount of fruit and vegetables, as suggested in the five-a-day recommendation, is not known. However, the observed effect of removing the free fruit program shows a decrease in fruit intake, but not vegetable intake for the intervention schools. This strengthens the validity of the findings since the program contributed very little to increase vegetable intake.

The reason for the decreased intake of FV may be caused by reduced availability and access of FV at the schools. A comprehensive review of children’s fruit and vegetable intake has identified availability and access as some of the strongest determinants of children’s and adolescents FV intake [39]. This is also supported by Norwegian findings [37,40,41].

In the present study, no third-order interactions were found, indicating that the effect of ending the program did not differ according to sex or parental education, which means that there was no difference between boys and girls, or between socioeconomic groups (based on parental education) in this sample. Thus, the findings do not confirm the results of studies showing that free school meals equalize social inequality [15,16,17], but also do not support the studies that show higher FV intake in high socioeconomic groups [12,13].

Several studies have shown an association between a higher FV intake and a lower intake of unhealthy snacks [30,31,42]. The results show that the relationship might also be reversed since pupils’ unhealthy snacking increased after ending the free fruit program. A significant increase in the intake of unhealthy snacks (+0.87 times/week) among participants at schools that ended the free fruit program was observed in the present study. In 2008, the intervention schools had a lower intake of unhealthy snacks compared to the control group (4.26 vs. 4.81, respectively), while in 2018, the results had shifted and pupils at the intervention schools had a higher intake of unhealthy snacks than the control schools (4.93 and 4.20, respectively).

To our knowledge, no previous studies have evaluated the effect of ending a free fruit program by using data from schools that previously received free fruit, but several follow-up studies have been conducted among children several years after the intervention period [43,44]. In the National Schools Fruit Scheme (NSFS) from England, the school-based fruit distribution appeared to have a short-term effect in increasing the FV intake of the pupils, but results were not obtained after the provision of free fruits ended [44].The FVMM project did an evaluation of the free fruit program three years after baseline and found that there might be a long-term effect of providing free fruit in school [45].

A strength of the current study is that it includes repeated data from a large number of randomly selected schools and an evaluation of the governmental initiative to end the free fruit program in a natural setting. The quasi-experimental design might, however, be seen as both a strength and a limitation, a strength because it gives the opportunity to evaluate the intervention in a real-life setting and a limitation because it often contributes to a bigger risk for bias and confounding [46]. Another limitation of this study is the non-randomization of the groups. However, no initial difference in parental education or sex were found between the groups (data not presented), indicating homogeneity in the sample. In addition, there were no significant difference in FV intake between the groups at baseline. However, the intervention schools reported eating unhealthy snacks 0.76 times/week less than the control schools in 2001 (*p* = 0.043).

The unhealthy snacks score is limited because it only included information about intake of three kinds of unhealthy snacks (candy, sugar-sweetened soft drink, and potato chips). However, the aim was not to present accurate data on the total intake of unhealthy snacks, but rather to investigate differences between the groups and time. 

It is also a limitation that three schools (one in the intervention group and two in the control group) had some kind of free fruit or free lunch paid by the municipalities after the free fruit program had ended. Therefore, some of the effects might be underestimated in the results, indicating that the effect could be even bigger than what was found in this study. In addition, most of the schools in the control group had a subscription program in 2018, while none of the intervention schools had the same. We have no data about how many of the pupils made use of the subscription program. 

There have been different project workers coding the 24 h recall at each test point, which makes it possible that the FV intake is interpreted differently each year. This makes it harder to compare the FV intake from 2018 to the intake in 2008 and 2001. Still, the main aim of the study was to evaluate the effect of ending the free fruit program by comparing the schools receiving free fruits with the control schools; thus, the possibly different interpretation of FV intake is of less importance. 

One should also be aware of the fact that there is a very long time between the measurement points (7 and 10 years). Thus, unknown factors, not registered in the present study, may have affected the FV intake during this time period.

### Areas for Further Research

There are no clear explanations as to why the pupils in the intervention group seem to eat less FV and more unhealthy snacks than the pupils in the control group. Other studies have suggested that longer lasting interventions [47] and a greater extent of involvement of parents [9,43,48] and teachers [9,49] are important to ensure positive intervention outcomes. Thus, future studies should examine the possible effects of involving role models and caretakers when implementing health-promoting intervention studies and evaluate whether longer intervention periods may result in sustainable effects.

## 5. Conclusions

Ending the free school fruit program implemented by the Norwegian government from 2007 to 2014, which provided a piece of fruit or vegetables every school day, decreased children’s intake of fruits and increased the consumption of unhealthy snacks. Ending the program seems to have a similar effect for all groups at school, regardless of sex and socioeconomic status. No effect was found on the intake of vegetables. Such evaluations of quasi-experiments are rare, but important in order to understand the effects of various health efforts implemented.

The authorities should reintroduce the free school fruit scheme, preferably together with better guidelines for how it should be implemented, as part of a public health strategy.

## Figures and Tables

**Figure 1 ijerph-20-02489-f001:**
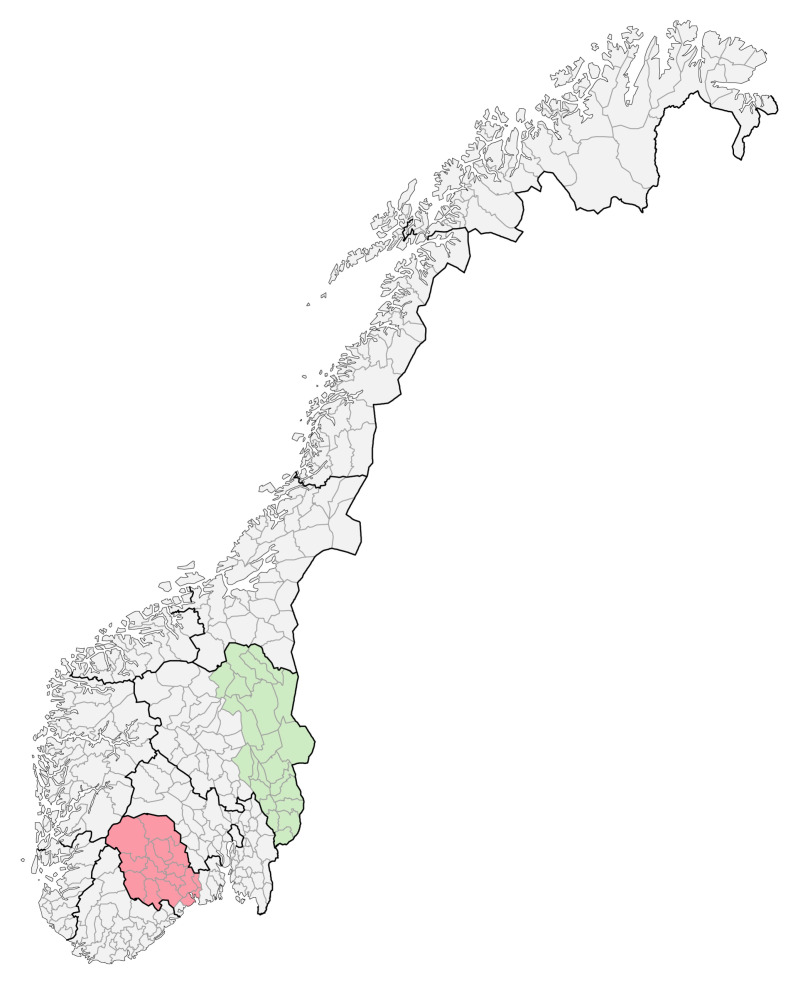
Telemark county in red, Hedmark in green. Hedmark county is now part of Innlandet county, and Telemark county is part of Vestfold and Telemark county after a county coalition in January 2020.

**Figure 2 ijerph-20-02489-f002:**
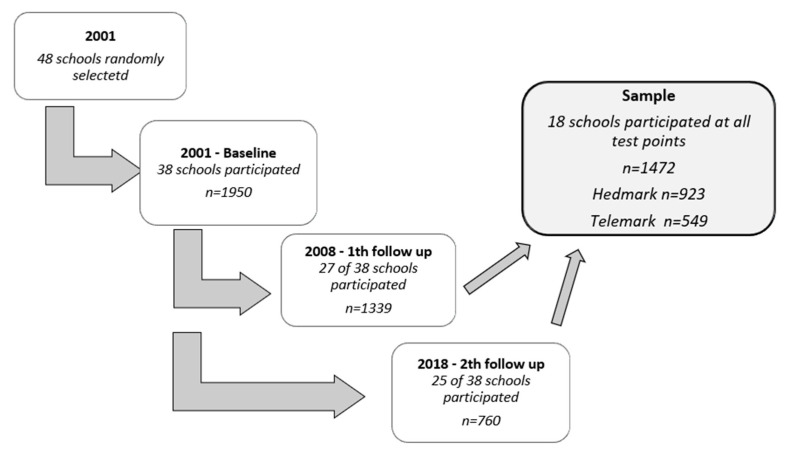
Study design, showing how many schools and pupils (*n*) are participating at each test point and in the present sample.

**Table 1 ijerph-20-02489-t001:** Description of the participants and the main variables in the 2001, 2008, and 2018 surveys.

	2001	2008	2018	*p* Value ^1^ 01–08	*p* Value ^1^ 08–18
**All**					
Number of schools	38	27	25		
Eligible pupils	2287	1712	1734		
Participating pupils	1950	1339	760		
Participation rate (%)	85	78	44		
**Sample**					
Number of schools	18	18	18		
**Pupil data**					
Participating pupils	963	911	561		
Sex, female (%)	49	52	53	0.258	0.064
Age, seventh graders (%)	48	49	53	0.777	0.094
FV intake all day (portions/d) ^2^	2.4 (±2.33)	3.2 (±2.8)	2.6 (±2.1)	<0.001 *	<0.001 *
FV intake at school (portions/d) ^2^	0.4 (±0.6)	0.8 (±0.9)	0.7 (±0.7)	<0.001 *	0.245
Eating FV 4–5 d/week at school (%)	28	65	56	<0.001 *	0.002 *
Unhealthy snacks (times/week) ^2^	6.9 (±4.3)	4.6 (±3.3)	4.3 (±2.8)	<0.001 *	0.016 *
**Parent data**					
Participating parents	809	668	431		
Participation rate (%)	84	73	77		
Sex, female (%)	85	79	79	0.006 *	0.988
Age (mean, years) ^2^	40.1 (±5.8)	40.9 (±5.1)	42.7 (±5.5)	0.004 *	<0.001 *
Higher education (%)	43	55	69	<0.001 *	<0.001 *

^1^ Based on t-test for continuous variables and the chi-square test for the dichotomous variables. ^2^ Mean ± SD. * Statistically significant, *p* < 0.05

**Table 2 ijerph-20-02489-t002:** Adjusted changes in fruit and vegetable (FV) intake at school and all day (portions/d), percentage of pupils eating FV 4 or 5 days/week, and unhealthy snacks (times/week) between 2008 to 2018 in relation to the school fruit program. (Mean values and 95% confidence intervals, CI).

	2008		2018		Change 08–18	*p* for Time × Group Interaction ^1^
	Mean	95% CI	Mean	95% CI		
**At school**						
*FV (portions/d)*						0.173
Intervention	0.82	0.57–1.06	0.53	0.22–0.85	−0.29	
Control	0.77	0.64–0.89	0.72	0.58–0.85	−0.05	
*Fruit (portion/d)*						**0.047**
Intervention	0.66	0.49–0.84	0.40	0.15–0.64	−0.27	
Control	0.51	0.42–0.60	0.52	0.42–0.62	0.01	
*Vegetables (portions/d)*						0.619
Intervention	0.16	0.03–0.28	0.14	−0.03–0.31	−0.02	
Control	0.26	0.19–0.32	0.20	0.12–0.27	−0.06	
**All day**						
*FV (portions/d)*						0.270
Intervention	3.26	2.54–3.98	2.19	1.25–3.12	−1,07	
Control	3.18	2.79–3.56	2.63	2.21–3.04	−0.55	
*Fruit (portion/d)*						0.128
Intervention	2.09	1.71–2.47	1.21	0.64–1.77	−0.88	
Control	1.97	1.77–2.17	1.60	1.37–1.82	−0.37	
*Vegetables (portions/d)*						0.902
Intervention	1.18	0.80–1.59	0.99	0.46–1.52	−0.19	
Control	1.19	0.97–1.41	1.03	0.80–1.27	−0.16	
**FFQ 4-5 times/week**						**<0.001**
Intervention (% yes)	85	70–100	50	32–69	−35	
Control (% yes)	61	53–69	59	50–68	−2	
**Unhealthy snacks**						
Intervention (times/week)	3.67	3.16–4.18	4.54	3.69–5.38	0.87	**0.012**
Control (times/week)	4.67	4.35–4.88	4.17	3.85–4.49	−0.50	

^1^ Based on multilevel mixed models adjusted for parental education, sex, and school. Numbers presented in bold text are statistically significant *p* < 0.05.

**Table 3 ijerph-20-02489-t003:** Proportion of pupils reporting to be eating fruits and vegetables at school 4 or 5 times/week stratified into groups (school fruit program in 2008) and sex/parental education level (percentage and 95% confidence intervals, CI).

		2001		2008		2018		Change 2001–08 p.p.	Change 2008–18 p.p.
	N	Percentage	95% CI	Percentage	95% CI	Percentage	95% CI		
**Intervention**									
Boys	191	15	6–23	79	70–88	38	22–54	64	−41
Girls	66	44	31–56	91	85–97	55	37–72	47	−36
Low parental education	44	31	20–43	80	69–91	57	27–87	48	−23
High parental education	140	30	12–42	91	85–98	41	24–59	62	−50
**Control**									
Boys	574	23	18–28	52	46–58	53	45–60	29	1
Girls	414	33	28–38	65	59–70	62	56–68	32	−3
Low parental education	307	29	24–34	55	49–62	57	48–67	27	2
High parental education	475	30	24–36	66	59–72	62	56–68	36	4

## Data Availability

Data available on request due to restrictions. The data presented in this study are available on request from elling.bere@uia.no. The data are not publicly available due to privacy reasons.

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
