# Peer review of "Effect of Ending the Nationwide Free School Fruit Scheme on the Intake of Fruits, Vegetables, and Unhealthy Snacks in Norwegian School Children Aged 10–12 Years"

_ijerph, 2023, doi:10.3390/ijerph20032489_

Round 1
Reviewer 1 Report
This is an interesting paper focusing on the effect of ending the nationwide free school fruit scheme in Norway on the intake of fruits, vegetables and unhealthy snacks. I have few suggestions/comments with the sole aim of improving the final version of this manuscript.
From the title, authors may need to include the category of school children (age-group).
However, from the abstract, some salient findings of this study are conspicuously missing.
There is a need to present a more robust background information on this study focusing on the keywords in the title. Give background view about fruits and vegetables consumption among schoolchildren and that of school feeding programs globally before focusing on that of Norway.
In the methodology, please you may add study area map. In line 112-157, you are expected to cite some previous empirical studies that have done similar work like the dietary diversity score, fruits/vegetables (FV) score. Add references(s) in line 127. Did you adapt any dietary diversity score approach for this study? If so, kindly cite here.
Before presenting Table 1, you are expected to give the results of the findings on FV consumption score among schoolchildren interviewed in this study.
Add reference(s) to the methodology involving unhealthy snacks consumption among schoolchildren in the study areas.
Your discussion from 239-262 should be supported by previous empirical studies.
Before the conclusion section, present a sub-section for "areas for further studies"
Present some recommendations enamating from this study immediately after the conclusion in a flowing sentences and not in bullet points or paragraphs. Thank you.
Author Response
Thank you very much for reviewing our manuscript. We have tried to respond to your comments as best we could. Please find attached our response to your points.

Author Response

(The authors gave the same response as above.)

Round 2
Reviewer 1 Report
This revised version showed that authors have made substantial changes as suggested in the original version of the manuscript. However, the Editor can now make final publication decision on this revised version. Thank you.